# Effects of Deposition and Annealing Temperature on the Structure and Optical Band Gap of MoS_2_ Films

**DOI:** 10.3390/ma13235515

**Published:** 2020-12-03

**Authors:** Gongying Chen, Benchu Lu, Xinyu Cui, Jianrong Xiao

**Affiliations:** 1College of Science, Guilin University of Technology, Guilin 541004, China; chengy1104@163.com (G.C.); benchuul@163.com (B.L.); cuixinyu0210@163.com (X.C.); 2School of Physics and Electronics, Central South University, Changsha 410083, China

**Keywords:** molybdenum disulfide films, magnetron sputtering, deposition temperature, crystal structure, optical band gap

## Abstract

In this study, molybdenum disulfide (MoS_2_) film samples were prepared at different temperatures and annealed through magnetron sputtering technology. The surface morphology, crystal structure, bonding structure, and optical properties of the samples were characterized and analyzed. The surface of the MoS_2_ films prepared by radio frequency magnetron sputtering is tightly coupled and well crystallized, the density of the films decreases, and their voids and grain size increase with the increase in deposition temperature. The higher the deposition temperature is, the more stable the MoS_2_ films deposited will be, and the 200 °C deposition temperature is an inflection point of the film stability. Annealing temperature affects the structure of the films, which is mainly related to sulfur and the growth mechanism of the films. Further research shows that the optical band gaps of the films deposited at different temperatures range from 0.92 eV to 1.15 eV, showing semiconductor bandgap characteristics. The optical band gap of the films deposited at 200 °C is slightly reduced after annealing in the range of 0.71–0.91 eV. After annealing, the optical band gap of the films decreases because of the two exciton peaks generated by the K point in the Brillouin zone of MoS_2_. The blue shift of the K point in the Brillouin zone causes a certain change in the optical band gap of the films.

## 1. Introduction

Molybdenum disulfide (MoS_2_) has a unique layered structure that is composed of S–Mo–S layers with covalent bonds and hexagonal coordination. Adjacent layers are formed by weak van der Waals forces [1,2]. MoS_2_ has many excellent properties and has been widely used in many fields, such as desulfurization [3,4], low consumption film transistors [5,6], and solid lubricants [7,8], because of its unique structure and is a 2D material with Dirac cone structure [9,10]. MoS_2_ film materials have excellent electrical and optical characteristics [11]. MoS_2_ film materials have a high carrier mobility, which is higher than that of organic silicon and amorphous silicon materials. The films are annealed or combined with other materials, such as silicon carbide crystals, thereby effectively improving their electrical properties [12,13,14,15]. The films have a direct band gap of 1.93 eV and an indirect band gap of 1.2 eV [16,17], and they are a promising photocatalyst [18]. Therefore, studying the structure and performance of MoS_2_ films is or great practical significance.

Many preparation methods are used for MoS_2_ film, such as the liquid phase peeling method, but it is difficult to control that the quantity and shape of the prepared MoS_2_ film [19,20]. Metal organic chemical vapor deposition can effectively prepare uniform and dense MoS_2_ films with high yield, but it requires high environmental conditions for preparation and the grain size is difficult to control [19,21]. Some researchers have used magnetron sputtering to prepare MoS_2_ films. This technology has good stability, high film forming purity, good uniformity and fast film formation speed [6,22,23,24,25], and magnetron sputtering has a large control range in terms of deposition temperature, substrate and kinetic energy of charged argon ions [5]. This is conducive to the preparation of films with various structures. However, the main disadvantage of magnetron sputtering is that vulcanization will lead to the growth of polycrystalline blocks, which makes it difficult to achieve single-layer film deposition [20,26]. Moreover, the film prepared by magnetron sputtering sometimes has the problem of high stress, but we can improve this problem by optimizing the experimental parameters and annealing. High-quality vertical structure and multiple active sites can be obtained through magnetron sputtering [27], and the MoS_2_ films have good photocatalytic performance [28,29]. A study found that the quality of the film can be improved with better catalytic performance after annealing and desulfurization [4,22,30]. At the same time, annealing has a great effect on improving the electrical properties of the films. After annealing, the MoS_2_ films have a Hall mobility of 4.40 cm^2^/Vs and a carrier concentration of 12.5 × 1016 cm^−3^. After annealing, the hysteresis voltage of the films can be expanded from 8.0 V to 28.4 V [31,32]. Heterostructures, such as Ni-MoS_2_ structure, can also be obtained through magnetron sputtering. These heterostructures have extremely fast optical response time, high cutoff frequency, high conductivity and high photocatalytic activity [24,25,33]. Few studies are conducted on changing the parameters in magnetron sputtering and studying the effects of the process parameters on the structure and optical properties of films.

In this work, magnetron sputtering technology was used to prepare MoS_2_ films at different deposition temperatures and annealed at different temperatures. The effects of deposition and annealing temperature on the crystal structure, surface morphology, and optical band gap of the MoS_2_ films were thoroughly studied.

## 2. Materials and Methods

MoS_2_ films were prepared on single crystal silicon (100) and quartz wafers using a JGP-450a magnetron sputtering equipment developed by Shenyang Scientific Instrument Co. (Shenyang, China), Ltd., Chinese Academy of Sciences. The silicon wafer or quartz wafer was immersed in an ethanol solution, washed with an ultrasonic cleaner for 15 min, rinsed with deionized water, and dried in an oven. The vacuum chamber was evacuated to 5.0 × 10^−4^ Pa. Argon gas (Ar, purity 99.99%) was introduced, and the Ar flow rate was stable at 30 standard cubic centimeters per minute (precisely controlled by a gas flow meter). The working air pressure was set to 1.0 Pa, and the RF power was 200 W. During sputtering, we used MoS_2_ target (Beijing Jingmai Zhongke Material Technology Co., Ltd., Beijing, China, Purity: 99.99%, Size: 60 × 3 mm, Tolerance: ±0.1 mm) and the deposition temperature was selected to be 50 °C, 100 °C, 200 °C, and 300 °C. The deposition time was 15 min (on the silicon substrate) and 3 min (on the quartz substrate). The MoS_2_ films prepared on the silicon and quartz wafers were annealed at different temperatures of 400 °C, 600 °C, and 800 °C. During annealing, argon gas (Ar, purity 99.99%) was continuously introduced. After reaching the annealing temperature, heating continued for 1 h and then naturally dropped to room temperature.

A scanning electron microscope (SEM, JSF-2100, Hitachi, Tokyo, Japan) was used to observe the surface morphology of MoS_2_ films on the silicon wafer before and after annealing, and various element contents of the films were obtained using an energy-dispersive spectrometer (EDS, Hitachi, Tokyo, Japan). An X-ray photoelectron spectrometer (XPS, Escalab250Xi, Thermo Fisher Scientific, Waltham, MA, USA) was used to analyze the bonding of various components of the thin film on the silicon wafer. The films on the silicon wafer was tested with an X-ray diffractometer (XRD, model X’pert3 Powder, Panalytical, Almelo, The Netherlands), and the physical image composition of the film was analyzed. An ultraviolet–visible light spectrometer (UV–vis, UV-2700, Shimadzu, Tokyo, Japan) was used to test the light transmission performance of the MoS_2_ films on the quartz wafer before and after annealing. In accordance with the light transmittance of the MoS_2_ films, their optical band gap was calculated using the Tauc equation.

## 3. Results and Discussion

### 3.1. Structure Analysis of the MoS_2_ Films

Figure 1a–d show the SEM surface morphology of the MoS_2_ films prepared at deposition temperatures of 50 °C, 100 °C, 200 °C, and 300 °C. Figure 1e shows a high-magnification SEM image of the films prepared at 300 °C, and Figure 1f shows the thickness of the film prepared at 200 °C (the thickness of the films is approximately 560 nm). The deposition temperature has a great influence on the surface morphology of the films. The MoS_2_ films deposited at 50 °C show a clear granular structure on the surface, as shown by the yellow circle in Figure 1a. The MoS_2_ films prepared under the conditions of 100 °C, 200 °C, and 300 °C show a “stripe” crystalline distribution (as shown by the yellow circle in Figure 1e), and the structural characteristics become evident with the increase in temperature. This structure is commonly found in the magnetron sputtering of MoS_2_ films, which can be attributed to their anisotropy. The growth rate of MoS_2_ at the edge of the vertical structure is different from the growth rate of the substrate, resulting in this intersecting structure. The surface of the MoS_2_ film becomes increasingly smooth and loose with the increase in deposition temperature, and many “feet” appear on the “stripe” from the SEM with high-magnification. The number of “feet” and the size increase with the increase in deposition temperature. These “feet” are the exposed edge sites of the MoS_2_ film, and these sites play an important role in improving the chemical activity and electron conduction [34,35]. A similar situation is found in the literature. When the deposition temperature is low, a granular surface morphology is easy to deposit, which is consistent with the SEM images observed in the literature [31]. A vertically aligned MoS_2_ films is easy to form with the increase in temperature [36,37]. This structure is because the growth rate of the films perpendicular to the substrate direction is slow, and the growth rate in the edge direction is slightly faster than that in the substrate direction at low temperatures. Therefore, this “particle” arrangement will appear on the film surface when the deposition temperature is low. The growth rate in the substrate direction increases, and the growth rate in the edge direction is relatively low when the temperature is high. Therefore, this dendritic structure will be formed on the film surface when the deposition temperature is high. The denser the surface of the films, the higher the deposition temperature and the looser they are when the surface structure of the films is “striped.” On this basis, the boundary point of the deposition temperature of the film surface morphology is 200 °C.

Figure 2 shows the SEM morphology of MoS_2_ films prepared at different deposition temperatures and subjected to different annealing temperatures. Figure 2a–d show the SEM images of the films annealed at 400 °C at different deposition temperatures. The films deposited at 50 °C and 100 °C showed a dense and smooth sheet structure after annealing at 400 °C. However, a large amount of filamentous structures appeared on the surface of the films deposited at 200 °C and 300 °C after annealing at 400 °C. The appearance of this filamentous structure may be caused by the migration of S in the MoS_2_ crystal during annealing. Therefore, the evident characteristics of the layered structure are retained, as shown in Figure 2c,d. Figure 2e–h show the SEM images of the films prepared at different deposition temperatures after annealing at 400 °C. The surface morphology of the annealed films in Figure 2e–g completely changes at 400 °C, and an evident layered structure is retained in Figure 2h. Annealing has a great influence on the MoS_2_ films and is affected by the deposition temperature. The surface structure of the film undergoes tremendous changes. This condition may be because of the difference in the mechanism of sulfur diffusion and the formation of MoS_2_ before and after annealing, thereby leading to changes in the morphology of the films. At the same time, high-temperature annealing can effectively improve the atomic mobility of the films, thereby allowing the atoms to be located in their energy matching position and improving the stability of the films [31]. Comparing the figures in Figure 2, the deposition temperature of 200 °C is the turning point of the surface structure of the films. The films prepared at 200 °C retain a good layered structure after annealing at 400 °C. The layered structure is also preserved at 300 °C, and the surface morphology of Figure 2a,b below the preparation temperature of 200 °C completely changes from that before annealing. Therefore, the films prepared at a deposition temperature of 200 °C have better stability, and the stability of the films is positively correlated with the deposition temperature, as shown in the comparison of the SEM images in Figure 2.

Figure 3 shows the EDS spectrum of the MoS_2_ films prepared at different deposition temperatures, and the insets show the contents of the corresponding elements. In accordance with the EDS data, the elemental ratios of S and Mo in the films are 1.86:1, 1.62:1, 1.55:1, and 1.18:1 at 50 °C, 100 °C, 200 °C, and 300 °C, respectively. The chemical ratio of molybdenum and sulfur in the actual films gradually decreases compared with the target chemical ratio with the increase in deposition temperature. This condition shows that the plasma energy increases with the increase in temperature, thereby destroying the S–Mo–S structure and forming a non-MoS_2_ chemical ratio structure. In Figure 3, a strong silicon element peak and an oxygen element peak appear. The silicon is attributed to the substrate single crystal silicon, and the oxygen element comes from the background atmosphere of the vacuum chamber.

Figure 4 shows the XRD pattern of MoS_2_ films prepared at different deposition temperatures. As shown in Figure 4a, each deposition temperature has a sharp peak at 62° corresponding to MoS_2_ (107), and a relatively weak peak corresponds to MoS_2_ (100) when the deposition temperature is higher than 50° at 33.3°. As shown in Figure 4b, two sharp peaks appear at 69.4° and 69.6° corresponding to MoS_2_ (202) and Si (100), respectively, the 33.3° peak corresponds to MoS_2_ (100), and the 61.9° peak corresponds to the MoS_2_ (107) plane, it is consistent with the XRD data in the literature [6,31,38]. Combined with the SEM results, the films are granular when the deposition temperature is low, and the growth rate is fast around this time. Thus, the crystallinity of the MoS_2_ (107) and MoS_2_ (202) surfaces is high. as the deposition temperature increases. The number of “feet” of the films gradually increases with the increase in deposition temperature when the film structure is “striped” to form a network structure. In particular, the number of “feet” is the largest at the temperature of 300 °C. Thus, the (107) and (202) faces of MoS_2_ have the highest crystallinity compared with 100 °C and 200 °C. This finding shows that the high deposition temperature in the “stripe” structure is beneficial to the growth of the lateral structure of the films. The MoS_2_ (100) surface only appears when the deposition temperature is high, which may be related to the growth of the films perpendicular to the substrate when the deposition temperature is high. These findings are in accordance with the SEM analysis of the MoS_2_ film surface structure changes with the deposition temperature.

Figure 5 shows the XRD pattern of the MoS_2_ films prepared at 200 °C then annealing at 400 °C, 600 °C, and 800 °C. As shown in Figure 5b, a strong peak appears at 62 °C after annealing for the MoS_2_ (107) plane, and the peak is more intense than before annealing. The peak value intensively changes with the change in annealing temperature. Figure 5a shows the small peaks of Mo and MoO_3_ [39]. This condition is because annealing desulfurization occurs after annealing, resulting in a small portion of MoO_3_ and Mo crystals. In accordance with the XRD spectrum, the grain size of the MoS_2_ films can be calculated using the Scherrer formula. The calculated results are shown in Figure 6. For the change in the grain size of the thin films at different deposition temperatures, the average grain size of the MoS_2_ films with a granular structure is larger than the average size of the “stripe” structure. This condition shows that the growth modes of the two different film structures are completely different. The film grain size of the “stripe”-like structure regularly increases, indicating that the higher the deposition temperature, the better the crystallization of the “stripe”-like film structure. In accordance with the film deposited at 200 °C, the grain size of the films with different annealing temperatures can be clearly observed with the rapid decrease in grain size and increase in annealing temperature, and the change rule of the grain size is in line with the phenomenon observed in Figure 2 that the surface morphology of the annealed film becomes loose as the annealing temperature increases. The films deposited at 200 °C have a large increase in grain size before and after annealing at 400 °C, indicating that a low annealing temperature is beneficial to the crystallization of the films. The films deposited at 200 °C are annealed at 600 °C or 800 °C, and the grain size is immensely reduced, thereby increasing the grain boundary area. This condition reduces the physical and chemical properties of the films, increases their impurities, and increases the film’s hardness and plastic toughness [40].

The XPS spectra of the MoS_2_ films prepared at 200 °C without annealing and 600 °C annealing are shown in Figure 7. In the deposition full spectrum, the binding energy of 230.0 and 161.5 eV correspond to Mo3d and S2p, respectively, as shown in Figure 7a. Two weak peaks are easily found at 35.0 and 37.0 eV, indicating the existence of the Mo4p valence state in the films. In the full spectrum after annealing at 600 °C, the S2p peak at 161.5 eV disappears, and a weak peak appears at 155 eV. The sharp O1s peak appears in the figure, and the Si2p peak appears at the binding energy of 106 eV. This condition shows that high-temperature annealing has a great influence on the valence of elements in the films. To analyze the chemical composition of the MoS_2_ films in depth, Lorentzian fitting was performed on the peaks in each of the two full spectra. The Mo3d peak before annealing can be fitted into two peaks, as shown in Figure 7a. The binding energy of 228.5 and 231.4 eV correspond to Mo3d_5/2_ and Mo3d_3/2_, respectively [41,42]. The presence of peaks at 232.1 and 234.8 eV can determine the presence of Mo^6+^ and indicate the presence of MoO_3_ in the film [41]. However, the peak area of Mo^6+^ is smaller than other peak areas, indicating that the main component of the films is still MoS_2_. The annealed Mo3d can still fit into a double peak, and the area of the Mo3d_5/2_ peak is immensely reduced. The MoO_x_ peak significantly increases. This condition shows that free molybdenum ions increase and the high temperature activates the oxygen on the film surface after high-temperature desulfurization. Therefore, the two react to form molybdenum oxide. The grain boundaries in the annealed grain size enlarge, and the film impurities increase. After annealing, the binding energy of each peak shifts in parallel and moves by 1.0 eV to the low-energy direction, making the structure of the films stable. The S2p high-resolution spectrum and fitting are shown in Figure 7b. Before annealing, the S2p_3/2_ and S2p_1/2_ peaks of the film are at the binding energy of 161.1 and 162.3 eV, respectively, thereby determining the presence of S^2−^ in the MoS_2_ films [43,44]. After high-temperature annealing, the S2p_3/2_ and S2p_1/2_ peaks immensely shift toward the binding, reducing the binding energy and making the structure stable. High-temperature desulfurization will improve the synthesis of MoS_2_ [45]. Figure 7c shows the high-resolution spectrum of O1s and the comparison between the two after fitting. The oxygen peak is activated under high-temperature conditions and produces many MoO_2_ impurities. Before and after annealing, the Si–O bonds with oxygen peaks at 531.58 and 533.9 eV originate from the SiO_2_ photoelectrons of the substrate [46].

### 3.2. Optical Band Gap of MoS_2_ Films

The transmission spectra of MoS_2_ films prepared at different deposition temperatures in a quartz substrate are shown in Figure 8a. In accordance with the transmission spectrum of Figure 8a combined with the SEM and XRD analyses of the MoS_2_ films, the film with high crystallinity has better transmittance. From the transmission curves of 100 °C, 200 °C, and 300 °C, the deposition temperature has a great influence on the film with “stripe” structure. Figure 8c shows the transmission spectrum of MoS_2_ films deposited at 200 °C with annealing at 400 °C, 600 °C, and 800 °C. As shown in Figure 2c, the surface structure of the films annealed at 200–400 °C is similar to that before annealing (see Figure 1c). The transmission curve is similar to the transmission curve before annealing but the transmission rate is reduced. The surface morphology of the films after annealing at 200–600 °C completely changes, resulting in an abrupt transmission curve. They are B and A exciton peaks at 200 nm to 350 nm and 650 nm to 850 nm, respectively. The two peaks are the transition regions of A and B excitons in the Brillouin zone and are caused by the blue shift of the K point in the Brillouin zone [16,47,48]. Therefore, a strong transmission peak appears at 400 nm to 650 nm. The spectral transmittance significantly increases with the increase in annealing temperature. This condition may be attributed to two reasons: First, the increase in annealing temperature will cause the spectral absorption rate of A and B excitons to decrease. Figure 8c shows that the annealing temperature will affect the K point in the Brillouin zone of MoS_2_. Second, annealing desulfurization changes the surface morphology and crystal structure of the film, resulting in a drop in its optical properties.

Combined with the transmission spectrum of the film, the optical band gap of the film can be calculated using the Tauc equation [49,50]:(1)$$\alpha =\frac{\ln(100/T)}{d}$$
(2)$${\left(\alpha \mathrm{hv}\right)}^{1/n}=A\left(\mathrm{hv}-E_{g}\right)$$
where α is the absorption coefficient, d is the film thickness, T is the transmittance, and A is a constant. n is determined by the specific situation of direct or indirect band gap, which is 0.5 or 2 [49]. The relationship between ${\left(\alpha hv\right)}^{1/2}$ and hv is calculated in accordance with the Tauc equation, and the tangent of the curve is obtained. The intersection of the tangent and the X axis is the optical band gap of the films. Figure 8b shows the optical band gap energy of MoS_2_ films at different deposition temperatures. Figure 8d shows the optical band gap of the MoS_2_ films prepared at 200 °C after annealing at different temperatures. Compared with the theoretical value (1.29 eV) in the literature [17], the band gap energy of films prepared at different deposition temperatures is reduced by 0.14–0.21 eV; the optical band gap energy of MoS_2_ films prepared at 200 °C is reduced by 0.32–0.58 eV. The deposition temperature and annealing treatment have a great influence on the band gap of the film. Combined with the SEM results, the optical band gap of the granular structure films is larger than that of the “stripe” structure, and the optical band gap of the “stripe” films is gradually widened with the increase in deposition temperature. Combined with the XRD analysis, the laterally grown crystal structure is beneficial to increase the band gap of the MoS_2_ films. Combined with the annealing temperature, the temperature has different effects on the growth mechanism of the MoS_2_ films, and that the films with granular and “stripe” structures are essentially different. Figure 8d shows that annealing has a great effect on the optical band gap of the MoS_2_ films. After annealing at 400 °C, the indirect optical band gap of the films is reduced by 0.06 eV compared with the as deposition films at 200 °C. The optical band gap changes slightly, indicating that the films change slightly. The band gaps of the films after annealing at 600 °C and 800 °C are 0.79 and 0.71 eV, respectively, indicating that high temperature annealing desulfurization will cause the crystal grain boundary to enlarge and affect the size of the optical band gap and the Brillouin zone. The K point has a great influence on the optical properties of the annealed MoS_2_ films.

## 4. Conclusions

At different deposition temperatures, the MoS_2_ film samples were prepared through magnetron sputtering and annealed. The results show that the surface of the thin film samples prepared by magnetron sputtering is flat and dense, with good quality and high crystallinity. The MoS_2_ films prepared at different deposition temperatures have different thermal stabilities. The higher the deposition temperature is, the better the thermal stability will be, and annealing desulfurization will occur. The suitable deposition temperature condition for preparing the MoS_2_ films is 200 °C. The deposition temperature has a great influence on the film growth mechanism, which is reflected in the change in surface morphology and crystal structure. The surface structure of the films changes from granular to vertically arranged “stripe” structure with the increase in deposition temperature. The density of the vertically aligned structure becomes loose with the increase in temperature, and the ratio of S and Mo decreases. The high deposition temperature is conducive to the lateral growth of the films. A part of Mo^6+^ is generated in the films when the deposition temperature is 200 °C, thereby forming MoO_3_ impurities and changing the composition and structure of the films. After annealing, the films have a tendency to agglomerate, and the lower the deposition temperature in the vertically aligned structure, the greater the strength of the film agglomeration will be. After high-temperature annealing, the average grain size of the films is greatly reduced, and the grain boundaries enlarge. At the same time, the impurities in the film increase after annealing, thereby changing its optical properties. However, annealing is beneficial to improve the structure of the films, which can increase their hardness and the toughness of their shape. After annealing at 600 °C (deposited at 200 °C), the binding energy of each peak of the films shifts toward the direction of structural stability. Studies have shown that the lateral growth of films has a great effect on light transmission. The indirect band gap of the films changes from 0.92 eV to 1.15 eV with the increase in deposition temperature. After annealing, the indirect band gap of the films prepared at 200 °C decreases. This condition is because of the increase in the grain boundary caused by annealing and the effect of the two exciton peaks on the optical band gap caused by the K-point blue shift of the MoS_2_ Brillouin zone. The study shows that the surface morphology, crystal structure, and composition of the MoS_2_ films can be improved by changing the deposition temperature and performing effective annealing. The results can be used to regulate the optical band gap of the films effectively and provide a reference for the application of MoS_2_ films in semiconductor optoelectronics and smart window films.

## Figures and Tables

**Figure 1 materials-13-05515-f001:**
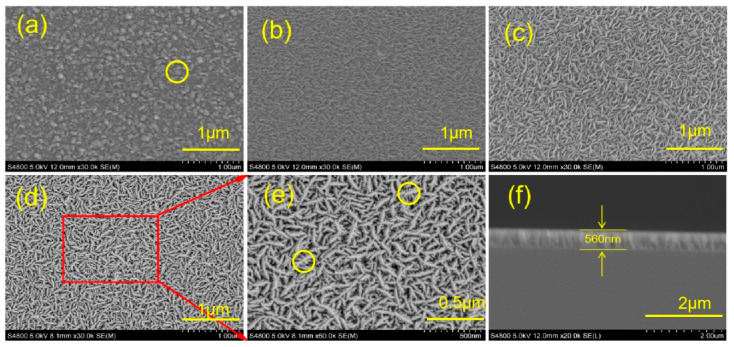
Scanning electron microscope (SEM) images of the MoS_2_ films prepared at different deposition temperatures: (**a**) 50 °C, (**b**) 100 °C, (**c**) 200 °C, (**d**) 300 °C, (**e**) high magnification, (**f**) cross section.

**Figure 2 materials-13-05515-f002:**
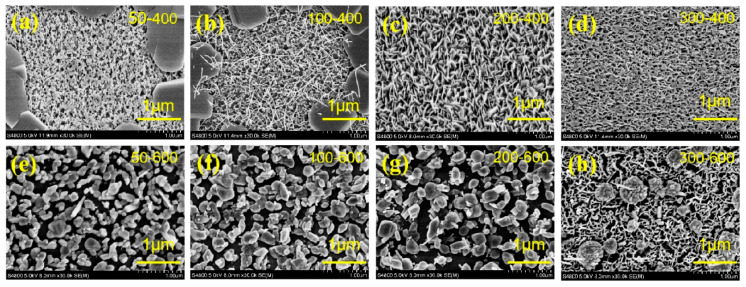
SEM surface morphology of the MoS_2_ films at different deposition–annealing temperatures: (**a**) 50–400 °C, (**b**) 100–400 °C, (**c**) 200–400 °C, (**d**) 300–400 °C, (**e**) 50–600 °C, (**f**)100–600 °C, (**g**) 200–600 °C, (**h**) 300–600 °C.

**Figure 3 materials-13-05515-f003:**
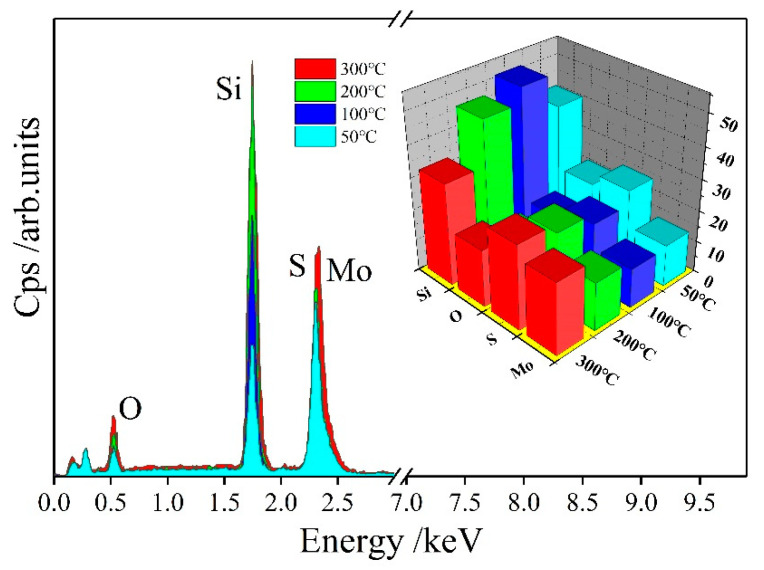
Energy-dispersive spectrometer (EDS) spectra of the MoS_2_ films prepared at different deposition temperatures.

**Figure 4 materials-13-05515-f004:**
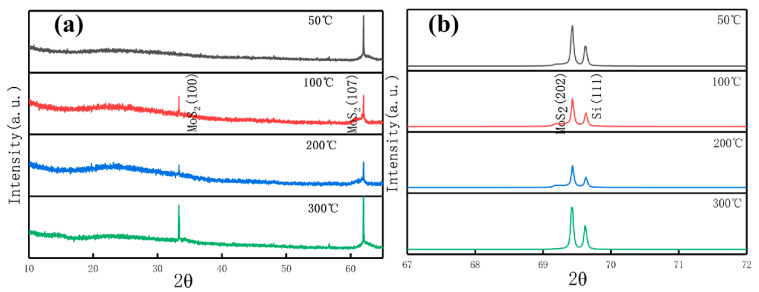
X-ray diffraction (XRD) patterns of the MoS_2_ films prepared at different deposition temperatures. Different diffraction angle range: (**a**) 2θ = 10°~65°, (**b**) 2θ = 67°~72°. (Picture a and b are continuous, and picture a is magnified).

**Figure 5 materials-13-05515-f005:**
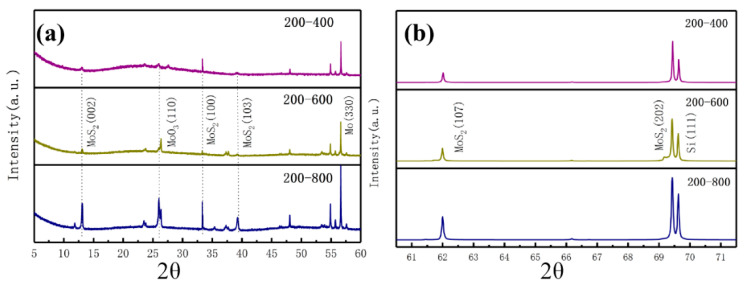
XRD spectrum of the MoS_2_ films prepared at 200 °C after annealing at 400 °C, 600 °C, and 800 °C. Different diffraction angle range: (**a**) 2θ = 5°~60°, (**b**) 2θ = 60.5°~71.5°. (Picture a and b are continuous, and picture a is magnified)

**Figure 6 materials-13-05515-f006:**
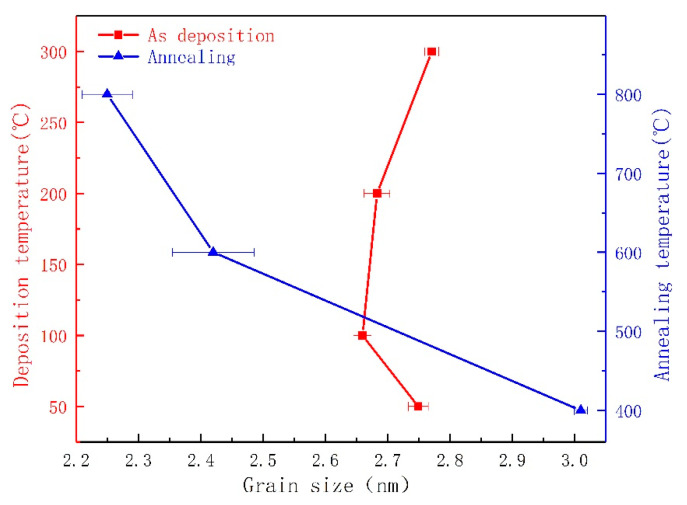
Relationship between the grain size of the MoS_2_ films and the deposition and annealing temperatures.

**Figure 7 materials-13-05515-f007:**
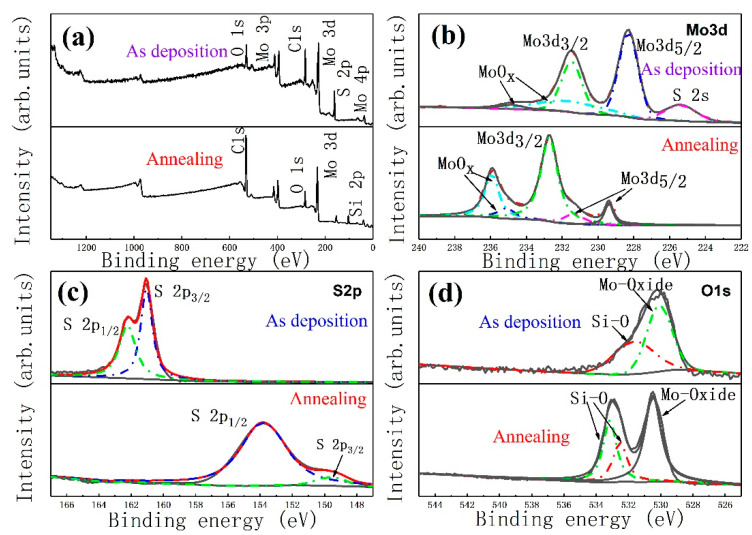
X-ray photoelectron spectrometer (XPS) pattern and Lorentzian fitting of each peak of the films deposited at 200 °C and the MoS_2_ films annealed at 200–600 °C. (**a**) total spectrum, (**b**) Mo3d, (**c**) S2p, (**d**) O1s.

**Figure 8 materials-13-05515-f008:**
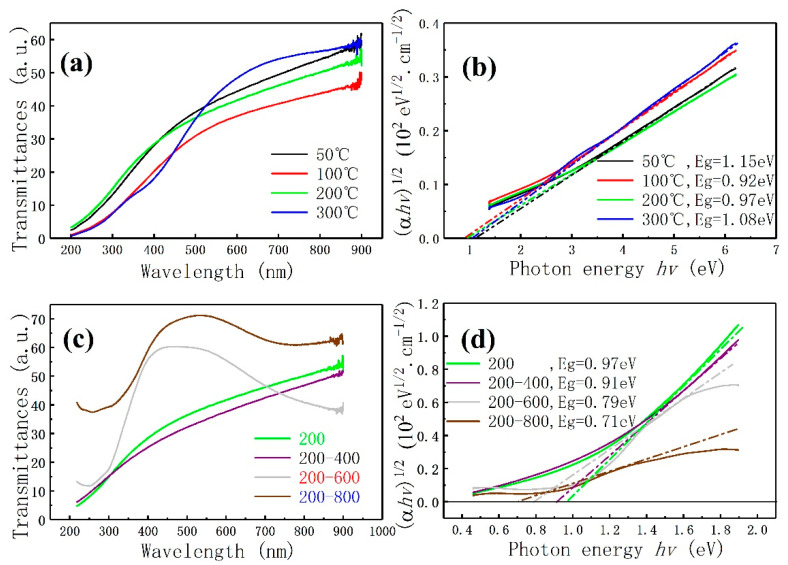
MoS_2_ films prepared at different deposition temperatures: (**a**) visible light transmittance, (**b**) the determination of the optical band gap. MoS_2_ films prepared at 200 °C annealing at different temperature: (**c**) visible light transmittance, (**d**) determination of optical band gap.

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
