# Peer review of "Effects of Deposition and Annealing Temperature on the Structure and Optical Band Gap of MoS2 Films"

_materials, 2020, doi:10.3390/ma13235515_

Round 1

Reviewer 1 Report

Reviewed paper can be published after come comments and corrections:

  1. Experimental section. What kind of target was used? There is no information.
  2. Fig. 1 and 2 (SEM). The scale bar is missing.
  3. Fig. 3. On a 3D plot, captions are illegible.
  4. Fig. 4 and 5. On a caption, better is use 2θ, that Ywo theta ame.
  5. Fig. 5. Why you presented data for samples deposited at 170C (as dep) not 100 and 200 C?
  6. XPS results. MoOx curve for Mo3d5/2 has to big FWHM. I think, you have to repeat XPS analysis.

Reviewer 2 Report

I cannot accept this manuscript for publication. I recommend that this manuscript be rewritten to verify the English style and that the structure of the session be re-orange, especially with regard to film morphology, heat treatment, and structure. The following points should be solved prior to a new submission elsewhere.

(1) Line 38: “Many preparation methods, such as mechanical peeling methods, are used for the MoS2 films” What the authors had in mind when speaking about “peeling methods”. In my opinion, this definition is not appropriate for MOCVD, or magnetron sputtering techniques.

(2) Since the manuscript presents results on MoS2 films obtained using the magnetron sputtering method, it would be useful to describe in more detail the main advantages and disadvantages of this technique in the section “Introduction”. Then it is necessary to compare the results obtained with literature data.

(3) Section 2 “Experiment”: In the first paragraph, the authors wrote that silicon (100) and quartz wafers were used as substrates for MoS2 film growth. What was the orientation of quartz substrates? Why further in the text (paragraph 2) the authors describe characterization methods only for films deposited on a silicon wafer. Also, it would be useful to explain the choice of substrate.

(4) Line 93: “The growth rate of the MoS2 films at the edge and the growth rate of the substrate are inconsistent, thereby leading to this intersecting structure“. Why did the authors compare the growth rate of the film at the edge and the wafer growth rate? Before publishing in the Journal, I strongly recommend to check carefully this manuscript and revise the English style.

(5) Figure 1e: The size and resolution of the image are not enough to see "foot" structures. For a better understanding, provide an image of these structures on a large scale.

(6) According to the caption to Figure 2, e-h images refer to MoS2 films that are annealed at 600 °C. Add appropriate comments to the text. Explain whether the various film samples were annealed at 400 °C and 600 °C, or the films were initially annealed at 400 °C and then re-annealed at 600 °C. Also, in the "Experiment" section, it is mentioned that a temperature treatment at 800 °C was carried out. Where are the relevant results and discussions?

(7) The spectra are shown in Figure 4 (b) are a continuation of the spectra shown in Figure 4 (a), aren't they? Or these spectra are measured under different conditions: (a) without annealing and (b) after thermal treatment? Why did the authors point that “two sharp peaks appear at 69.4° and 69.6° corresponding to MoS2 (202) and Si (100), respectively, the 33.3° peak corresponds to MoS2(100), and the 61.9° peak corresponds to the MoS2(107) plane”.

(8) If the authors also used quartz wafers for MoS2 growth, then where are the results and discussion in this regard?

Reviewer 3 Report

The authors have shown in this manuscript that the surface morphology, crystal structure, and composition of the MoS2 films can be improved by changing the deposition temperature and performing effective annealing. This work is interesting, however, there are a few things that are not clear:

  • The authors have mentioned in the introduction that the grain size is difficult to control, and in this manuscript, the grain size is calculated but it is still not clear why the grain size is important in this manuscript.
  • The scale bar in Figure 1 and 2 are too small to read. Can the authors describe how the cross-sectional image were prepared. Can the authors also indicated “foot”, “particle” and “striped” structures on SEM images?
  • Most of the figures in this work show poor resolution.
  • The authors have described the XRD patterns but there is no need of showing formula 1 to 3 in this work as it is common known formula. It is sometimes difficult to see the results, therefore, the authors are suggested to design a table overview of the obtained results.
  • Can the authors give the error bar of the calculated grain size?

Round 2

Reviewer 1 Report

After corrections paper can be published.

Reviewer 2 Report

The authors responded to the comments made, significantly improving and supplementing the document according to some suggestions. Thus, the manuscript can be published in a journal.